# *Acinetobacter baumannii* Infections in Hospitalized Patients, Treatment Outcomes

**DOI:** 10.3390/antibiotics10060630

**Published:** 2021-05-25

**Authors:** Diaa Alrahmany, Ahmed F. Omar, Gehan Harb, Wasim S. El Nekidy, Islam M. Ghazi

**Affiliations:** 1Pharmacy Department, Sohar Hospital, Sohar 311, Oman; diaa.alrahmany@yahoo.com; 2General Medicine Department, Sohar Hospital, Sohar 311, Oman; Farouqmail2000@yahoo.com; 3Gehan Harb Statistics, Cairo 11511, Egypt; gehan.harb74@gmail.com; 4Cleveland Clinic Abu Dhabi, Abu-Dhabi P.O. Box 112412, United Arab Emirates; ElnekiW@ClevelandClinicAbuDhabi.ae; 5Cleveland Clinic Lerner College of Medicine, Case Western Reserve University, Cleveland, OH 44195, USA; 6Philadelphia College of Pharmacy, University of the Sciences, Philadelphia, PA 19104, USA

**Keywords:** *Acinetobacter baumannii*, mortality, treatment outcomes, antimicrobial, resistance, adverse events, recurrence, LOS, treatment regimens, colistin, tigecycline

## Abstract

*Background Acinetobacter baumannii* (AB), an opportunistic pathogen, could develop into serious infections with high mortality and financial burden. The debate surrounding the selection of effective antibiotic treatment necessitates studies to define the optimal approach. This study aims to compare the clinical outcomes of commonly used treatment regimens in hospitalized patients with AB infections to guide stewardship efforts. *Material and methods*: Ethical approval was obtained, 320 adult patients with confirmed AB infections admitted to our tertiary care facility within two years were enrolled. The treatment outcomes were statistically analyzed to study the relation between antibiotic regimens and 14, 28, and 90-day mortality as the primary outcomes using binary logistic regression—using R software—in addition to the length of hospitalization, adverse events due to antibiotic treatment, and 90-day recurrence as secondary outcomes. *Results*: Among 320 patients, 142 (44%) had respiratory tract, 105 (33%) soft tissue, 42 (13%) urinary tract, 22 (7%) bacte iemia, and other infections 9 (3%). Nosocomial infections were 190 (59%) versus community-acquired. Monotherapy was significantly associated with lower 28-day (*p* < 0.05, OR:0.6] and 90-day (*p* < 0.05, OR:0.4) mortality rates, shorter length of stay LOS (*p* < 0.05, Median: −12 days] and limited development of adverse events (*p* < 0.05, OR:0.4). Subgroup analysis revealed similar results ranging from lower odds of mortality, adverse events, and shorter LOS to statistically significant correlation to monotherapy. Meropenem (MEM) and piperacillin/tazobactam (PIP/TAZ) monotherapies showed non-significant high odd ratios of mortalities, adverse events, and disparate LOS. There was a statistical correlation between most combined therapies and adverse events, and longer LOS. Colistin based and colistin/meropenem (CST/MEM) combinations were superior in terms of 14-day mortality (*p* = 0.05, OR:0.4) and (*p* < 0.05, OR:0.4) respectively. Pip/Taz and MEM-based combined therapies were associated with statistically non-significant high odd ratios of mortalities. Tigecycline (TGC)-based combinations showed a significant correlation to mortalities (*p* < 0.05, OR:2.5)*. Conclusion*: Monotherapy was associated with lower mortality rates, shorter LOS, and limited development of adverse events compared to combined therapies. Colistin monotherapy, colistin/meropenem, and other colistin combinations showed almost equivalent mortality outcomes. Patients on combined therapy were more susceptible to adverse events and comparable LOS. The possible adverse outcomes of PIP/TAZ and MEM-based therapies in the treatment of MDRAB infections and the association of TGC with a higher mortality rate raise doubts about their treatment role.

## 1. Introduction

*Acinetobacter baumannii* is a widespread opportunistic pathogen that colonizes human skin and the respiratory tract [1]. It could develop into serious infections such as septicaemias [2], endocarditis [3], pneumonia [4], meningitis [5], and wound infections [6]. Senility, immunocompromised status, prolonged hospitalization, invasive procedures, exposure to broad-spectrum antimicrobials, and admissions to critical care areas are among the most common risk factors for *A. baumannii* hospital-acquired infections [7,8,9].

Treatment failure, prolonged hospitalizations, and high mortality rates among hospitalized patients due to multiple drug-resistant *Acinetobacter baumannii* (MDRAB) remain a cumbersome challenge to health care providers [10,11]. The growing antimicrobial resistance of *A. baumannii* prompted the Centers for Disease Control and Prevention (CDC) in 2013 [12] and the World Health Organization (WHO) in 2017 to highlight carbapenem-resistant *A. baumannii* (CRAB) as a serious threat and high priority target for research and development of new antibiotics [13].

*A. baumannii* possesses one of the most extensive genetic resistance islands, comprising multiple resistance genes, denoting the tremendous capability of *A. baumannii* to resist harsh environmental conditions and acquire resistance determinants under antibiotic pressure. Deactivating enzymes production, overexpressing efflux pumps, and biofilm formation grant the ability for colonization and nosocomial spread [14]. The limited therapeutic options caused by the increased *A. baumannii* resistance to almost all relevant antimicrobials [15,16,17,18,19], the mounting resistance to carbapenem and colistin caused by the abusive prescription was associated with treatment failures [20], prolonged hospitalization [21], increased healthcare-related costs [22] and high mortality rates [23].

The scarcity of available antimicrobial options due to the growing resistance and lack of novel antibiotics led to the increased prescribing of last-resort agents such as colistin and tigecycline to target MDRAB. This excessive prescribing will eventually accelerate the resistance rates to these drugs [24,25].

Consequent to this mounting antimicrobial resistance of *A. baumannii*, no optimized-treatment regimens have been defined for multidrug-resistant *A. baumannii* (MDRAB) infections yet. Therefore, the recommendation of using monotherapy [26] or combination therapy [27] should be based on multiple studies evaluating the effect of each of the two strategies on treatment outcomes. Optimal lab testing, regular reviews, and protocolization of treatment options to augment clinical outcomes are of paramount importance.

In a previous surveillance study by our group, “exploring bacterial resistance in northern Oman” [28], a high mortality rate associated with *A. baumannii* infections was observed. In the present work, we attempted to investigate the clinical outcomes of various antibiotic regimens commonly prescribed in our hospital to treat *A. baumannii* infections as a stewardship initiative to guide the proper selection of antimicrobial therapy.

## 2. Methods

This retrospective study included adult patients (>18 years) with a confirmed *A. baumannii* infection who were admitted to our tertiary care facility (Suhar Hospital, Oman) within 2 years (1 January 2016 to 31 December 2017). Patient-relevant data were collected from the hospital’s electronic medical records after the study was approved by the Research and Ethical Review Committee, North Batinah Governorate, Ministry of Health, Sultanate of Oman.

We examined the patients’ demographics; age, gender, clinical sign and symptoms of infection (to exclude patients with colonization), underlying comorbid conditions (diabetes mellitus, chronic renal failure, active malignancy, immuno-suppressed, chronic cardiac diseases, chronic Respiratory disease), 90-day exposure to invasive procedures, 90-day prior hospitalization history, and prior infections. Hospitalization details; admission diagnosis, discharge status, length of hospitalization, admission ward, infection acquisition place, readmission rates. Microbiological details; laboratory-confirmed identification of *A. baumannii* in samples collected from infection sites, specimen type, susceptibility pattern, resistance phenotype, concomitant infections. For patients with ≥2 positive cultures, only the first episode was selected. Patients with positive *A. baumannii* culture who had not been admitted, who died before receiving one dose of antibiotics, and pediatric patients were excluded.

### 2.1. Definitions

Hospital-acquisition: infection occurred > 48 h of the admission date; all other episodes were considered community-acquired infections [29]. Critical care stays: admission to intensive care unit (ICU), cardiac care unit (CCU), or burn unit (BU) for more than 24 h. At the end of treatment, the clinical prognosis was defined as a complete or partial resolution of signs of infection, normal laboratory values (WBC, CRP), or negative culture of the same source of the original infection. Mortality was considered if the symptomatic patient had a positive *A. baumannii* culture and death occurred before the resolution of signs of *A. baumannii* infection during the same hospitalization at days 14 and 28 of admission, 90-day was all-cause mortality. Adverse events: untoward clinical occurrence likely related to the use of antibiotics as described in the literature. Acute kidney failure (deranged estimated glomerular filtration eGFR at the beginning of treatment), development of fungal infections (proven by microbiological culture), or *Clostridioides difficile* infection (proven microbiologically) that may present during treatment with antibiotics [30]. The treatment of *A. baumannii* using a single antibiotic is considered monotherapy, while combined therapy is the use of 2 or more antibiotics with antimicrobial effect towards *A. baumannii* during the infection episode. Statistically analyzed antibiotic regimens were selected based on commonly prescribed antibiotics with activity against *A. baumannii*.

MDRAB was defined following CLSI 2010 M100-S20 guidance [31] as the *A. baumannii* isolate resistant to at least one antibiotic of three or more antimicrobial classes. Carbapenem resistant *A. baumannii* CRAB was phenotypically detected—according to CLSI—as the isolates that showed inhibition zones < 23 mm around (ertapenem 10 μg or meropenem 10 μg), and tested-resistant to one or more agents in cephalosporin subclass III (e.g., cefotaxime, ceftazidime, and ceftriaxone). Confirmed CRAB was reported as resistant for all penicillins, cephalosporins, carbapenems, and aztreonam.

### 2.2. Statistical Analysis

We studied the statistical relation between antibiotic regimens and 14, 28, and 90-day mortality as the primary outcomes, in addition to three secondary outcomes; length of hospital stays (LOS), adverse events due to antibiotic treatment, and 90-day recurrence. The study cohort was studied as a whole and then divided into subgroups depending on the source of infection (hospital vs. acquired), infection site (bacteremia, respiratory, soft tissue, and urine), and organism phenotype (MDR versus sensitive).

The data were analyzed using R software statistical programming language, version 3.6.2 (2019-12-12) “Dark and Stormy Night” (R Foundation for Statistical Computing platform). The numeric data were described as median and interquartile range and analyzed using linear regression analysis after the normality (tested with Shapiro–Wilk normality test). Categorical data is analyzed using binary logistic regression and expressed using *p* values, odds ratios, and confidence intervals. The impact of treatment regimens (mono versus combined, colistin (CST), meropenem (MEM), piperacillin/tazobactam (PIP/TAZ), ceftazidime (CAZ), and tigecycline (TGC)-based) on 14, 28, and 90 days mortality, length of hospital stay (LOS), adverse events, and 90-day recurrence were analyzed using binary logistic regression.

All tests were two-sided, and *p*-values < 0.05 are considered significant with 95% confidence level). The odds ratio is the ratio of the odds of study outcomes in the presence/absence of each of the studied antimicrobial treatment regimens.

## 3. Results

### 3.1. Patient Characteristics

A total of 320 patients were included in this study for two years. Of these, 180 patients (56%) were males, and 140 (44%) were females. The median age of the study cohort was 63 years (IQR 39–75), with a median for survivors 48 years (31–70) and expired patients 69 years (61–77). Among the 320 patients, 142 (44%) had respiratory tract infections, 105 (33%) skin and soft tissue infections, 42 (13%) urinary tract, 22(7%) bacteremia, and 9 (3%) suffered from other infections. Nosocomial infections were 190 (59%), and 41% were community-acquired (CAIs). Patients admitted to critical care areas were 92 (29%).

### 3.2. Microbiological Features

Of the total infection episodes, 260 (81%) were caused by MDRAB isolates, the majority of isolates were susceptible to colistin CST (99%) and tigecycline TGC (86%), medium susceptibility to doxycycline DOX (48%), and co-trimoxazole SMX/TMP (37%), poor susceptibility to all remaining antibiotics: amikacin AMK (25%), ciprofloxacin CIP (17%), ceftazidime CAZ (18%), gentamycin GEN (19%), meropenem MEM (17%), and piperacillin/tazobactam PIP/TAZ (17%). Table 1 shows the susceptibility pattern of all *A. baumannii* isolates and antibiotics use metrics from different sample sources.

### 3.3. Antimicrobial Treatment

The antimicrobial regimens during infection episode were as follows: CST-based 162 (51%), MEM-based 100 (31%), PIP/TAZ 89 (28%), and TGC-based 37 (12%). According to antimicrobial regimens, the cohort is divided into two comparable groups; 175 (55%) of patients received combined therapy while 145 (45%) received monotherapy. Table 2 shows the demographics and the details of antibiotics regimens per sample source, and Table 1 shows the susceptibility pattern and antibiotic use metrics vs. sample source.

### 3.4. Outcomes

None of the antimicrobial regimens had any statistically significant effect on 90-day recurrence rates, possibly due to the low number of recurrences 8 (3%). Adverse events related to prolonged use of antimicrobials occurred in 46% of the patients; nephrotoxicity 6%, development of fungal infections 93%, or skin rashes and *Clostridium difficile* (both 1%).

Monotherapy in general was significantly associated with lower 28-day (*p* < 0.05, OR: 0.62, CI: (0.38, 1.0)) and 90-day (*p* < 0.001, OR: 0.46, CI: (0.29, 0.73)) mortality rates, shorter LOS (*p* < 0.01, Median: −12 days), and limited development of antibiotic-related adverse events (*p* < 0.001, OR: 0.47, CI: (0.30, 0.74)) compared to patients received combined therapy. Subgroup analysis revealed similar results ranging from non-significant lower odds ratios of mortality, adverse events, and shorter LOS to high statistically significant positive correlation to monotherapy. Except for MEM and PIP/TAZ monotherapies which showed non-statistically significant high odd ratios of all types of mortalities, adverse events, and disparate LOS among subgroups. See Table 3 for more detailed odd ratios and *p* values.

In the overall study sample, as well as in subgroups analysis, CST-based and CST/MEM combined therapy were superior in terms of 14-day mortality rate (*p* < 0.01, OR: 0.47, CI: (0.26, 0.84)) and (*p* < 0.05, OR:0.47, CI: (0.21, 1.05)) respectively. That was not the case with other combinations that showed results ranging from relatively high odd ratios of mortalities with PIP/TAZ and MEM-based combinations to statistically significant impact with TGC-based (*p* < 0.03, OR: 2.3, CI: (1.12, 4.77)). There was a statistical correlation between most of the combined therapies and the occurrence of adverse events, with longer LOS when studying the whole sample and subgroups. See Table 3.

## 4. Discussion

Poor clinical outcomes are related to infections caused by CRAB, MDRAB, and (Extensive drug-resistant (*A. baumannii*) XDRAB due to the scarcity of available effective antimicrobial treatment and the growing resistance to currently available options. We studied and assessed the impact of many antimicrobial regimens on 14, 28, and 90-day mortality, the occurrence of adverse events, LOS, and 90-day recurrence to guide physicians’ judgment and provide opportunities for antimicrobial stewardship de-escalation strategy. The patients in each regimen group were compared to the patients in the rest of the cohort.

### 4.1. Monotherapy vs. Combined

Generally, in our work, monotherapy was significantly associated with lower mortality rates and shorter LOS compared to combined therapy, which may be explained by the limited development of antibiotic-related adverse events (22–46%) in the patients who received monotherapy compared to 60% in those receiving combined therapies. Subgroup analysis revealed similar findings ranging from non-significant lower odds ratios of mortality, adverse events, and shorter LOS to a high statistically significant impact of monotherapy (see Table 3).

### 4.2. Colistin

CST monotherapy was associated with significantly fewer 14 and 28-day deaths, especially in MDR-related infections (*p* < 0.03, OR: 0.35, CI: (0.12, 1.02)) and HAI’s (*p* < 0.01, OR:0.14, CI: (0.02, 1.03)), meanwhile, it was related to fewer deaths in other subgroups. Patients treated with CST monotherapy were less likely to experience adverse events. However, they needed longer LOS compared to their counterparts who underwent other monotherapies.

CST combined therapy had superior outcomes only in the case of 14-day mortality (*p* < 0.008, OR: 0.47, CI: (0.26, 0.84)), with disparate non-significant high odd ratios in 28 and 90-day mortalities. Patients receiving CST combined therapy were more susceptible to adverse events than those who received CST monotherapy and comparable LOS. While many studies reported a non-significant synergism of CST combinations to reduce mortality [27,32,33,34,35], per contra; others reported higher 14-day mortality with CST monotherapy compared to CST-based combinations [36,37]. This dissimilarity may be driven by the patient’s conditions, the existence of multiple infections, and the initiation time of colistin treatment.

CST/MEM combination and CST monotherapy showed equivalent 14-day mortality outcome (*p* < 0.047 Vs 0.049, OR: 0.38 Vs 0.48) respectively. CST/MEM combinations showed an almost similar effect on 28 and 90-day mortalities, adverse events, and LOS compared to other CST combinations. The non-superiority of CST/MEM over CST monotherapy was proven by Shi [34], Yilmaz [27], and others, while antagonized by Park [36], Katip [38] and Jiaying Li [39], and others [40], which draw the attention to the impact of site of infection and the type of carbapenemases on the treatment outcomes.

### 4.3. Ceftazidime (CAZ)

Statistically, significantly fewer deaths, shorter LOS, and noticeably fewer adverse events were associated in patients treated by CAZ monotherapy (31/320). The effect on mortality was more obvious in 28- and 90-day mortality (*p* < 0.04, OR: 0.38, CI: (0.14, 1.02)) and (*p* < 0.002, OR: 0.26, CI: (0.10, 0.68)) respectively. In the same context, many studies found CAZ an effective empiric treatment of Gram-negative-caused febrile neutropenia in cancer patients [41,42]. In our study, CAZ was mostly prescribed in simple monomicrobial CAI’s (19/31) 61%, where the causative organism was mostly sensitive phenotype which explains the good outcomes. CAZ is used in combination in few cases (17/320); therefore, it was difficult to monitor any exact statistical value.

### 4.4. Meropenem

MEM monotherapy was associated with non-statistically significant high odds of mortalities among most subgroups. The high prevalence of MDRAB-related infections among the whole sample (81%) and the decreased susceptibility to meropenem (17%) may explain this high prevalence of mortality. These patients have shorter LOS and are less likely to experience adverse events, which may be due to the high safety profile of carbapenems compared to other antimicrobials and the high (early death)14-day mortality rate associated with this regimen (33%).

MEM combined therapy was generally characterized with non-significant longer LOS with high odds of 14- and 28-day mortalities and significant contributions to 90-day mortality (*p* < 0.006, OR: 2.02, CI: (1.22, 3.34) and adverse events (*p* < 0.04, OR: 1.67, CI: (1.01, 2.75)). Even though MEM induces phenotype divergence together with carbapenem resistance [43], many recent studies recommended MEM monotherapy as a promising treatment for MDR bacterial isolates provided that it is administered as an optimized two-step-administration therapy (OTAT) but not as the currently traditional simple prolonged-infusion (TSPI) [44].

### 4.5. Piperacillin/Tazobactam

Only 17% of study isolates were susceptible to Pip/Taz, among the 28% of the patients treated by PIP/TAZ-based therapy (mono- and combined therapy). High odds of adverse events and poor mortality outcomes were noticed, especially in MDRAB-related infections, with variable LOS among subgroups. Many studies reported doubtful invitro bactericidal effect of β-lactam/β-lactamases inhibitors combinations on MDRAB compared to non-*A. baumannii* strains which were more susceptible to these combinations [45]. Ampicillin/sulbactam is the exception that appeared in many studies as an efficacious treatment option for CRAB strains and significantly decreased the risk of death [46,47,48], which may be explained by the intrinsic whole-cell antimicrobial activity of sulbactam in addition to the enhanced β-lactam availability [49].

### 4.6. Tigecycline

Although 84% of the isolates were susceptible to TGC, TGC-based therapy was associated with highly significant odd ratios of 14, 28, 90-day mortalities, adverse events, and prolonged stays, among all subgroups. These undesired outcomes were quite similar to many studies [50,51] that led the US Food and Drug Administration (FDA) in 2010 to issue a warning declaration to practitioners that TGC is linked to higher mortality rates than with comparator drugs, especially in ventilator-associated pneumonia and bacteremias [52]. In 2013 FDA approved a new Boxed warning about this risk to be added to the Tygacil^®^ drug label and updated the Warnings and precautions and the Adverse reactions sections. A possible explanation of TGC-based treatment-related mortality is TGC bacteriostatic action and the inferior availability of the drug in the blood, which might lead to secondary bacteremias.

Our study included a convenient number of patients (320) over a long duration (two years); the sample included all types of *A. baumannii* infections that occurred within the study period, which offered good generalisability of study results. We also tried to monitor the main outputs related to treatment outcomes (mortalities, adverse events, LOS, and recurrence).

The present study had some limitations being a single-centered, retrospective study. The coexistence of other concurrent infections and underlying comorbid conditions with *A. baumannii* infection in the study cohort was challenging, particularly for interpreting the clinical outcome and mortality. We found that age, case severity, or the number of underlying diseases did not confound the relationship between treatment regimens and treatment outcomes. They are almost equally distributed among both groups (mono and combined therapy). See Table 4. Concurrent infections did not have any statistically significant confounding effect on mortality related to the failure of antimicrobial treatment as shown by regression analysis, 14-day mortality (*p* = 0.873, OR: 0.95, CI: (0.52, 1.73)), 28-day mortality (*p* = 0.293, OR: 1.35, CI: (0.77, 2.36)).

## 5. Conclusions

Generally, monotherapy was significantly associated with lower mortality rates, shorter LOS, and limited exposure to antibiotic-related adverse events compared to combined therapies. Despite the patient’s conditions, compared with CST monotherapy, CST/MEM and other CST combinations showed equivalent primary mortality outcomes in treating MDRAB. Patients on combined therapy were more susceptible to adverse events and comparable LOS.

CAZ remains an optimal option for empiric treatment of *A. baumannii* CAI’s and simple monomicrobial infections. Meanwhile, further prospective studies are needed to investigate the effect of modified (OTAT vs. TSPI) regimens of MEM against MDRAB and XDRAB isolates.

The doubtful effect of PIP/TAZ-based therapy in the treatment of CRAB excludes the benefit of using β-lactam/β-lactamases inhibitors in the treatment of serious infections caused by *A. baumannii*. Meanwhile, the association of TGC with high mortality rates includes that TGC cannot be relied on in the treatment of *A. baumannii* serious infections.

## Figures and Tables

**Table 1 antibiotics-10-00630-t001:** Susceptibility pattern and antibiotic use metrics.

Susceptibility Pattern (tested)	Blood *n* = 22	Respiratory *n* = 142	Skin and Soft Tissue *n* = 105	Urine *n* = 42	Others *n* = 9	Average Susceptibility.	Average DDD	Average DOT
Amikacin (*n* = 68)	4 (18%)	24 (17%)	25 (24%)	13 (33%)	2 (25%)	(24%)	9.4	5.6
Ciprofloxacin (*n* = 54)	4 (18%)	22 (16%)	18 (18%)	9 (22%)	1 (11%)	(17%)	16.0	7.1
Colistin (*n* = 282)	21 (100%)	131 (98%)	92 (99%)	30 (100%)	8 (100%)	(99%)	2.2	9.4
Co-trimoxazole (*n* = 112)	3 (14%)	39 (28%)	46 (45%)	22 (56%)	2 (22%)	(33%)	6.8	7.2
Ceftazidime (*n* = 57)	3 (14%)	21 (15%)	21 (20%)	11 (26%)	1 (11%)	(17%)	3.3	6.0
Gentamycin (*n* = 58)	3 (15%)	21 (16%)	21 (21%)	11 (27%)	2 (22%)	(20%)	29.5	4.5
Meropenem (*n* = 52)	3 (14%)	16 (12%)	18 (19%)	15 (41%)	0 (0%)	(17%)	6.4	7.8
Piperacillin/Tazobactam (*n* = 58)	3 (14%)	22 (15%)	22 (21%)	10 (24%)	1 (11%)	(17%)	1.3	7.0
Tigecycline (*n* = 180)	13 (87%)	90 (85%)	55 (89%)	19 (86%)	3 (75%)	(84%)	142.3	8.0
Doxycycline (*n* = 73)	6 (43%)	25 (40%)	25 (50%)	17 (77%)	0 (0%)	(42%)	90.0	5.7

Others: Body fluids and patient-related deceives; DDD: Defined daily dose; DOT: Days of therapy; Average susceptibility = Sum of susceptibility percentages/5.

**Table 2 antibiotics-10-00630-t002:** Patients’ characteristics.

Variable	Blood No. (%)	Respiratory No. (%)	Skin and Soft Tissue No. (%)	Urine No. (%)	Others No. (%)
Total (*n* = 320)	22 (7%)	142 (44%)	105 (33%)	42 (13%)	9 (3%)
Male Gender (*n* = 180)	15 (68%)	92 (65%)	59 (56%)	8 (19%)	6 (67%)
Admission to Critical Care (*n* = 92)	9 (41%)	59 (42%)	18 (17%)	3 (7%)	3 (33%)
Resistance (MDR) (*n* = 260)	18 (82%)	120 (85%)	84 (80%)	30 (71%)	8 (89%)
Hospital Acquired (*n* = 190)	15 (68%)	92 (65%)	55 (52%)	23 (55%)	5 (56%)
90-day recurrence (*n* = 8)	1 (5%)	3 (2%)	4 (4%)	0 (0%)	0 (0%)
Adverse Event (*n* = 147)	9 (41%)	93 (65%)	37 (35%)	7 (17%)	1 (11%)
Combined Therapy (*n* = 175)	14 (64%)	92 (65%)	49 (47%)	13 (31%)	7 (78%)
Pip/Taz Monotherapy (*n* = 35)	4 (18%)	16 (11%)	12 (11%)	3 (7%)	0 (0%)
Pip/Taz Combined Therapy (*n* = 54)	4 (18%)	29 (20%)	16 (15%)	2 (5%)	3 (33%)
Pip/Taz based (*n* = 89)	8 (36%)	45 (32%)	28 (27%)	5 (12%)	3 (33%)
CST Monotherapy (*n* = 37)	1 (5%)	11 (8%)	18 (17%)	7 (17%)	0 (0%)
CST Combined Therapy (*n* = 125)	8 (36%)	70 (49%)	35 (33%)	8 (19%)	4 (44%)
CST based (*n* = 162)	9 (41%)	81 (57%)	53 (50%)	15 (36%)	4 (44%)
CAZ Monotherapy (*n* = 31)	1 (5%)	11 (8%)	9 (9%)	9 (21%)	1 (11%)
MEM Monotherapy (*n* = 15)	0 (0%)	4 (3%)	6 (6%)	4 (10%)	1 (11%)
MEM Combined Therapy (*n* = 85)	8 (36%)	46 (32%)	22 (21%)	6 (14%)	3 (33%)
MEM based (*n* = 100)	8 (36%)	50 (35%)	28 (27%)	10 (24%)	4 (44%)
Other Monotherapies (*n* = 27)	2 (9%)	8 (6%)	11 (10%)	6 (14%)	0 (0%)
TGC based Therapy (*n* = 37)	4 (18%)	25 (18%)	6 (6%)	0 (0%)	2 (22%)
CST + MEM based (*n* = 59)	4 (18%)	36 (25%)	15 (14%)	3 (7%)	1 (11%)
Polymicrobial Infections (*n* = 239)	17 (77%)	115 (81%)	79 (75%)	21 (50%)	7 (78%)
≥ 3 comorbidities (*n* = 127)	8 (36%)	58 (41%)	49 (47%)	9 (21%)	3 (33%)

MDR: Multiple Drug Resistence; Pip/Taz: Piperacillin/Tazobactam; CST: Colistin; CAZ: Ceftazidime; MEM: Meropenem; TGC: Tigecycline.

**Table 3 antibiotics-10-00630-t003:** Treatment outcomes for *A. baumannii* infections.

	14 Days Mortality					28 Days Mortality					90 Days Mortality					Adverse Events					LOS				
	*p* Value (Odd Ratios)					*p* Value (Odd Ratios)					*p* Value (Odd Ratios)					*p* Value (Odd Ratios)					± Median (*p* Value)				
	Total	Hospital	Community	Sensitive	MDR	Total	Hospital	Community	Sensitive	MDR	Total	Hospital	Community	Sensitive	MDR	Total	Hospital	Community	Sensitive	MDR	Total	Hospital	Community	Sensitive	MDR
Monotherapy	0.983	0.795	0.817	0.736	0.677	0.047	0.795	0.140	0.879	0.155	0.001	0.028	0.027	0.484	0.018	0.001	0.280	0.002	0.047	0.055	−12	−3	−8	−8	−9
	(0.99)	(0.91)	(1.10)	(0.77)	(1.13)	(0.62)	(0.91)	(0.57)	(0.89)	(0.68)	(0.46)	(0.50)	(0.44)	(0.61)	(0.54)	(0.47)	(0.72)	(0.32)	(0.28)	(0.61)	(0.01)	(0.98)	(0.02)	(0.18)	(0.12)
Pip/Taz monotherapy	0.015	0.079	0.088	0.882	0.005	0.148	0.079	0.268	1.000	0.061	0.492	0.593	0.489	0.888	0.230	0.295	0.902	0.048	0.899	0.138	−9	−16.5	−2	7.5	−12
	(2.55)	(2.82)	(2.43)	(1.14)	(3.42)	(1.71)	(2.82)	(1.73)	(1.00)	(2.22)	(1.28)	(1.35)	(1.40)	(0.89)	(1.66)	(1.46)	(0.93)	(2.60)	(1.10)	(1.89)	(0.09)	(0.33)	(0.39)	(0.64)	(0.13)
CST monotherapy	0.047	0.010	0.876	***	0.031	0.061	0.010	0.749	***	0.024	0.290	0.182	0.971	***	0.103	0.157	0.202	0.448	***	0.049	17	11	4	***	14
	(0.38)	(0.14)	(0.90)	***	(0.35)	(0.46)	(0.14)	(0.82)	***	(0.39)	(0.68)	(0.54)	(0.98)	***	(0.55)	(0.60)	(0.56)	(0.63)	***	(0.49)	(0.30)	(0.18)	(0.954)	***	(0.13)
CAZ monotherapy	0.143	0.362	0.226	0.615	0.988	0.036	0.362	0.105	0.335	0.981	0.000	0.031	0.044	0.459	0.980	0.001	0.064	0.030	0.485	0.007	−14	−16	−7	−5	−16
	(0.47)	(0.41)	(0.47)	(1.43)	(0.00)	(0.38)	(0.41)	(0.41)	(1.92)	(0.00)	(0.26)	(0.15)	(0.34)	(1.64)	(0.00)	(0.25)	(0.25)	(0.31)	(0.64)	(0.17)	(0.20)	(0.53)	(0.28)	(0.93)	(0.45)
MEM monotherapy	0.341	0.709	0.343	0.910	0.168	0.902	0.709	0.749	0.826	0.535	0.568	0.417	0.935	0.747	0.992	0.116	0.764	***	0.483	0.304	−13	−12	−4	−1.5	−16
	(1.74)	(1.39)	(2.11)	(0.88)	(2.66)	(1.07)	(1.39)	(1.28)	(0.78)	(1.54)	(0.73)	(0.52)	(1.06)	(0.70)	(0.99)	(0.41)	(1.26)	***	(0.48)	(0.49)	(0.33)	(0.65)	(0.52)	(0.45)	(0.58)
Other monotherapies	0.939	0.381	0.315	0.282	0.392	0.247	0.381	0.074	0.222	0.914	0.038	0.472	0.036	0.174	0.359	0.329	0.275	0.027	0.296	0.790	−13	−13	−6	−3.5	−13
	(0.96)	(1.78)	(0.48)	(0.35)	(1.65)	(0.59)	(1.78)	(0.29)	(0.31)	(0.94)	(0.40)	(0.64)	(0.24)	(0.27)	(0.60)	(0.67)	(1.96)	(0.22)	(0.44)	(1.15)	(0.13)	(0.36)	(0.40)	(0.60)	(0.28)
CST combined	0.008	0.038	0.062	0.731	0.001	0.439	0.038	0.185	0.801	0.976	0.003	0.072	0.033	0.194	0.079	0.049	0.779	0.053	0.343	0.384	18	11	9	25	15
	(0.47)	(0.48)	(0.33)	(1.53)	(0.38)	(1.21)	(0.48)	(1.80)	(1.36)	(1.01)	(1.99)	(1.70)	(2.52)	(4.09)	(1.55)	(1.57)	(1.09)	(2.31)	(2.73)	(1.24)	(0.17)	(0.61)	(0.82)	(0.28)	(0.60)
MEM combined	0.629	0.895	0.310	0.479	0.899	0.064	0.895	0.035	0.562	0.158	0.006	0.264	0.003	0.177	0.050	0.044	0.623	0.020	0.088	0.279	7	4	8.5	11	3.5
	(1.16)	(0.95)	(1.66)	(1.96)	(1.04)	(1.64)	(0.95)	(2.59)	(1.72)	(1.49)	(2.02)	(1.42)	(3.77)	(3.23)	(1.70)	(1.67)	(1.17)	(2.81)	(4.10)	(1.34)	(0.97)	(0.41)	(0.78)	(0.55)	(0.61)
Pip/Taz combined	0.809	0.561	0.719	0.994	0.760	0.377	0.561	0.197	0.994	0.444	0.328	0.901	0.168	0.994	0.481	0.954	0.180	0.087	0.846	0.639	1.5	−3	6	−2	−1
	(1.09)	(1.29)	(0.81)	(0.00)	(1.12)	(1.32)	(1.29)	(1.91)	(0.00)	(1.28)	(1.34)	(1.05)	(1.97)	(0.00)	(1.25)	(1.02)	(0.60)	(2.32)	(1.28)	(0.86)	(0.05)	(0.72)	(0.00)	(0.04)	(0.19)
TGC combined	0.015	0.002	0.807	***	0.022	0.000	0.002	0.061	***	0.000	0.000	0.000	0.023	***	0.000	0.000	0.002	0.000	***	0.000	13	0	13.5	***	10
	(2.55)	(3.99)	(0.82)	***	(2.46)	(5.79)	(3.99)	(3.50)	***	(5.26)	(7.29)	(8.95)	(4.67)	***	(6.23)	(6.84)	(4.46)	(18.0)	***	(5.93)	(0.02)	(0.56)	(0.00)	***	(0.06)
CST based therapy	0.000	0.000	0.090	0.731	0.000	0.694	0.000	0.333	0.801	0.130	0.027	0.328	0.062	0.194	0.521	0.304	0.568	0.222	0.343	0.611	22	17.5	8	25	19.5
	(0.36)	(0.28)	(0.44)	(1.53)	(0.25)	(0.91)	(0.28)	(1.47)	(1.36)	(0.67)	(1.66)	(1.35)	(2.05)	(4.09)	(1.18)	(1.26)	(0.84)	(1.60)	(2.73)	(0.88)	(0.04)	(0.15)	(0.81)	(0.27)	(0.34)
MEM based	0.363	0.982	0.145	0.625	0.899	0.068	0.982	0.035	0.757	0.158	0.018	0.440	0.006	0.382	0.058	0.221	0.550	0.371	0.370	0.511	3.5	2.5	4.5	5	1
	(1.29)	(1.01)	(1.94)	(1.46)	(1.04)	(1.59)	(1.01)	(2.39)	(1.27)	(1.49)	(1.79)	(1.27)	(3.08)	(1.88)	(1.65)	(1.34)	(1.20)	(1.44)	(1.82)	(1.19)	(0.63)	(0.32)	(0.92)	(0.93)	(0.48)
Pip/Taz based	0.051	0.107	0.262	0.752	0.760	0.079	0.107	0.058	0.630	0.444	0.194	0.666	0.103	0.522	0.163	0.436	0.209	0.004	0.827	0.585	−4	−10.5	3	3	−8
	(1.75)	(1.85)	(1.63)	(0.77)	(1.12)	(1.59)	(1.85)	(2.12)	(0.67)	(1.28)	(1.39)	(1.16)	(1.88)	(0.60)	(1.47)	(1.22)	(0.66)	(3.10)	(1.16)	(1.16)	(0.65)	(0.36)	(0.09)	(0.52)	(0.86)
TGC based	0.028	0.006	0.807	***	0.039	0.000	0.006	0.061	***	0.000	0.000	0.000	0.023	***	0.000	0.000	0.003	0.000	***	0.000	13	1	12.5	***	10
	(2.31)	(3.41)	(0.82)	***	(2.22)	(4.85)	(3.41)	(3.50)	***	(4.39)	(5.61)	(5.85)	(4.67)	***	(4.77)	(6.08)	(3.91)	(18.0)	***	(5.27)	(0.02)	(0.50)	(0.00)	***	(0.08)
CST+ Mem based	0.049	0.181	0.095	0.731	0.022	0.501	0.181	0.381	0.801	0.760	0.035	0.219	0.090	0.194	0.174	0.157	0.943	0.033	0.343	0.447	18	5	12	25	15
	(0.47)	(0.56)	(0.23)	(1.53)	(0.40)	(1.23)	(0.56)	(1.67)	(1.36)	(1.10)	(1.84)	(1.52)	(2.63)	(4.09)	(1.51)	(1.51)	(0.98)	(3.42)	(2.73)	(1.26)	(0.67)	(0.53)	(0.49)	(0.27)	(0.95)

Pip/Taz: Pipracillin/Tazobactam; CST: Colistin; CAZ: Ceftazidime; MEM: Meropenem; TGC: Tigecycline; MDR: Multi-drug resistant; Pink background: Significant negative; blue background: Significant positive. *** The software is unable to calculate *p* value due to insufficient frequency of the variable.

**Table 4 antibiotics-10-00630-t004:** Age and comorbidities in comparison groups.

Comorbid Condition	Combined Therapy (*n* = 175)		Monotherapy (*n* = 145)		*p* Value
No of comorbidities Median (IQR)	2 (1–3)		2 (0-3)		0.29
Age Median (IQR)	62.72 (39.2–73.2)		62.84 (38.5–76.3)		0.80
Diabetes mellites	77	44%	54	37%	0.22
Chronic renal failure	36	21%	22	15%	0.21
Active malignancy	7	4%	7	5%	0.72
Immuno-suppressed	4	2%	5	3%	0.53
Chronic Cardiac Diseases	100	57%	78	54%	0.55
Chronic Resp. Disease	20	11%	15	10%	0.76

## Data Availability

Raw data are archived in Sohar Hospital, North Batinah Governorate, Sultanate of Oman, electronic data are protected in H drive of RGU Citrix work place. Data sharing is pending approval from medicolegal authorities.

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
