# Peer review of "Acinetobacter baumannii Infections in Hospitalized Patients, Treatment Outcomes"

_antibiotics, 2021, doi:10.3390/antibiotics10060630_

Round 1
Reviewer 1 Report
This is an important paper that examines the relationship between treatment and prognosis of Acinetobacter infections, but as the authors indicate, there are several limitations.
Major:
- Patients who receive combination antimicrobial therapy may be more severely ill or have a critical background. For example, Acinetobacter infections are sometimes problems in severe burns, and it is not uncommon in my experience to use multidrug therapy in burn patients. However, the prognosis for these patients is poor, not related to Acinetobacter. I recommend adding more information in the limitations about the possible impact of differences in background between combination therapy and monotherapy group.
  2. I'm sorry, but I didn't know what Figure 1. was showing.
Minor:
3. Line 138-139
Did the difference in age groups between survivors and non-survivors affect the results of this study?
Author Response
Dear reviewer,
Please attached a detailed point-by-point response to the comments.
Much appreciated,
Corresponding author

Reviewer 2 Report
The manuscript by Alrahmany et al. presents a retrospective study evaluating the efficacy of antibiotic treatments for 320 patients against A. baumannii. This is of high importance for clinical applications the success of which relies on the therapeutic efficacy.
My main concern is that the manuscript is written as a highly abbreviated clinical report. It is very difficult to follow the results. Besides, the style is cumbersome with long convoluted sentences. Some sentences are just lists with no appropriate grammatical structure.
The methods are too abbreviated. The data analysis was not described, just a list of tools is provided, which is not sufficient. On L. 92-96, the authors just list characteristics without explaining. Organizing this information into visual charts or tables may serve better.
Figure 1 is difficult to follow and requires a better explanation. The legend has to spell out all the abbreviations. Need to provide the information about the total number of isolates from each source and in each group of antibiotic resistance. Were there any antibiotic-sensitive isolates? When describing susceptibility, need to explain how the “high”, “medium”, and “poor” were assigned. I do not know how to relate the numbers in fig 1 and the numbers provided in 3.2.
In 3.3, do the numbers represent patients? Tables are difficult to follow. It is not clear what these numbers exactly represent. For table 3, is there a way to present the data graphically and illustrate the correlations?
The discussion is also difficult to follow. Why the patients receiving monotherapies had less exposure to “antibiotic-related adverse events”? The authors mention “subgroups” but never explain what they are.
The authors relate the mortality and other factors to different therapies but did not explain what was the basis for these therapies to be administered in the first place. It may be important to consider both the patient status determining the therapy and the therapy of choice when analyzing the outcome.
Other comments
- L. 24 Spell out LOS in the abstract
- L.31 Sone information is missing here. Just one “other combined therapy”?
- L. 36 “equal”?
- L. 55 Edit the sentence
- L.68-76 Such sentences are way too long and convoluted.
- L. 91 what does it mean “to exclude colonization”?
- L. 97 What 2 positive cultures? “were”
- L. 98 If the patients have not been admitted, why would they be considered as hospital patients?
- Need to define what 14, 28, and 90-day mortality is and how these numbers relate to mortality rate. Define what “odd ratios” are.
- L. 164 What is “the sample”?
Author Response
Dear reviewer,
Please see a point-by-point response to the comments.
Much appreciated,
Corresponding author

Round 2
Reviewer 1 Report
I checked the changes made by the authors and considered that appropriate corrections have been made to the parts I pointed out.
